# Somatic Reversion of a Novel *IL2RG* Mutation Resulting in Atypical X-Linked Combined Immunodeficiency

**DOI:** 10.3390/genes13010035

**Published:** 2021-12-23

**Authors:** Yujuan Hou, Hans Peter Gratz, Guillermo Ureña-Bailén, Paul G. Gratz, Karin Schilbach-Stückle, Tina Renno, Derya Güngör, Daniel A. Mader, Elke Malenke, Justin S. Antony, Rupert Handgretinger, Markus Mezger

**Affiliations:** Department of Pediatrics I, Hematology and Oncology, University Children’s Hospital, University of Tübingen, 72076 Tübingen, Germany; Yujuan.Hou@med.uni-tuebingen.de (Y.H.); Hans-Peter.Gratz@med.uni-tuebingen.de (H.P.G.); Guillermo.Urena@med.uni-tuebingen.de (G.U.-B.); Paul.Gratz@med.uni-tuebingen.de (P.G.G.); Karin.Schilbach@med.uni-tuebingen.de (K.S.-S.); tina.renno@student.uni-tuebingen.de (T.R.); Derya.Guengoer@med.uni-tuebingen.de (D.G.); Daniel.Mader@med.uni-tuebingen.de (D.A.M.); Elke.Malenke@med.uni-tuebingen.de (E.M.); Justin.Selvaraj@med.uni-tuebingen.de (J.S.A.); Rupert.Handgretinger@med.uni-tuebingen.de (R.H.)

**Keywords:** atypical X-SCID, immunodeficiency, IL-2RG, γ_C_, somatic reversion, hypomorphic mutation

## Abstract

Mutations of the *IL2RG* gene, which encodes for the interleukin-2 receptor common gamma chain (γ_C_, CD132), can lead to X-linked severe combined immunodeficiency (X-SCID) associated with a T^−^B^+^NK^−^ phenotype as a result of dysfunctional γ_C_-JAK3-STAT5 signaling. Lately, hypomorphic mutations of the *IL2RG* gene have been described causing atypical SCID with a milder phenotype. Here, we report three brothers with low-normal lymphocyte counts and susceptibility to recurrent respiratory infections and cutaneous warts. The clinical presentation combined with dysgammaglobulinemia suspected an inherited immunity disorder, which has been proven by Next Generation Sequencing as a novel c.458T > C; p.Ile153Thr *IL2RG* missense-mutation. Subsequent functional characterization revealed impaired T-cell proliferation, low TREC levels and a skewed TCR Vβ repertoire in all three patients. Interestingly, investigation of various subpopulations showed normal expression of CD132 but with partially impaired STAT5 phosphorylation compared to healthy controls. Additionally, we performed precise genetic analysis of subpopulations revealing spontaneous somatic reversion, predominately in lymphoid derived CD3^+^, CD4^+^ and CD8^+^ T cells. Our data demonstrate that the atypical SCID phenotype noticed in these three brothers is due to the combination of hypomorphic IL-2RG function and somatic reversion.

## 1. Introduction

X-linked severe combined immunodeficiency (X-SCID) is an inherited and life-threatening primary immunodeficiency disorder (PID) that is caused by mutations in the interleukin-2 receptor gamma chain gene (*IL2RG*) [1]. The gene is located on the X-chromosome q13.1 and it encodes for the common gamma chain (γ_C_) which is a subunit of various interleukin receptors, such as IL-2, IL-4, IL-7, IL-9, IL-15, and IL-21 and constitutes the high-affinity complex with the α and β chains complementing the IL-2 receptor. IL-2R gamma chain works as a signaling kinase that is responsible for signal transduction of the common gamma chain of cytokine receptor subfamilies by cooperating with other JAK and STAT proteins [2,3,4]. The common gamma chain plays an important role in the proliferation and differentiation of lymphocytes, therefore, mutations in the *IL2RG* gene inevitably cause a lack of T cells and natural killer- (NK) cells, and non-functional B lymphocytes [5]. To date, more than 200 mutations in the *IL2RG* gene have been identified [6]. The majority of them are single base substitutions (missense and nonsense mutations), followed by splice site mutations, deletion, and insertion mutations [6]. The mutations can result in non-functional γ_C_ or the production of a stop codon which prevents lymphocyte development. Thus, in typical X-SCID, the lack of IL-2RG function leads to diverse defects in humoral and cellular immunity. Patients are highly susceptible to bacterial and viral infections showing various symptoms like chronic diarrhea, skin rashes, and the failure to thrive. Additionally, B-cell dysfunction and hypogammaglobulinemia are commonly found [7]. Unless the immune system is reconstituted through bone marrow transplantation or gene therapy approaches, children die within the first two years of life [8].

Diverse mutations in the *IL2RG* gene are implicated in different phenotypic variants. Therefore, patients can present with atypical X-SCID whose symptoms range from normal to moderately reduced T and NK cell frequency and function resulting in a “milder” form of immunodeficiency. Approximately 10% of the reported *IL2RG* mutations have been associated with atypical and highly variable immune phenotypes that are described as hypomorphic mutations, which impair but do not abrogate protein function, obscure and mitigate clinical presentation of the disease [6,9]. Additionally, a few cases of atypical phenotype with prolonged survival are reported because of spontaneous somatic reversion of the mutations [10]. Somatic reversion involves the correction of a pathogenic mutation in cells that is a naturally occurring phenomenon so that it is described as “natural gene therapy” [11]. Multiple reversion events result in somatic mosaicism characterized by partial or complete reversion of mutated somatic cells to a wild-type allele resulting in two genetically distinct populations of cells within an individual [12,13,14]. Lately, an increasing number of studies have identified revertant somatic mosaicism in patients affected with PIDs, in which the co-existence of mutant and revertant cells has been described [14]. To the best of our knowledge, only a few cases of X-SCID have been reported with revertant mutations in the *IL2RG* gene revealing somatic mosaicism [10,15,16,17,18,19,20,21,22,23]. All of these patients have a significant attenuation of the clinical phenotype since there are variable reverted mutations within lymphocyte sub-populations.

In this study, we describe the clinical and molecular performance of a novel *IL2RG* single nucleotide variant c.458T > C; p.Ile153Thr in three 20-plus-year-old brothers who show mild symptoms of immunodeficiency. We characterized the revertant somatic mosaicism in subset lymphocytes derived from the three patients in terms of proliferation and function.

## 2. Materials and Methods

### 2.1. Patient Recruitment 

The patients were recruited through the outpatient clinic of the Center of General Pediatrics, Oncology/Hematology, at the University Children’s Hospital, Tübingen. Clinical data were collected retrospectively. Informed consent for this study was obtained in accordance with the Declaration of Helsinki and the Institutional Review Board approval from the University of Tübingen Ethics Committee (No. 928/2020BO2).

### 2.2. Samples

Heparinized peripheral blood was obtained from the three patients, their mother, and healthy donors at a similar age as the patients. Peripheral blood mononuclear cells (PBMCs) were isolated from peripheral blood by density gradient centrifugation using Biocoll^®^ Trennlösung (BIO&SELL, Nuremberg, Germany) according to the manufacturer’s instructions. CD3^+^ T cells were collected from PBMCs with the magnetic-activated cell sorting system (CliniMACS System, Miltenyi Biotec, Bergisch Gladbach, Germany) based on the manufacturer’s instructions of MACS CD3 Microbeads and LS separation columns. The purity of gathered CD3^+^ T cells determined by flow cytometry (BD FACS Calibur, BD Biosciences, Franklin Lakes, NJ, USA) was >90% for all experiments (data not shown). Various subsets of CD3^+^, CD4^+^, CD8^+^, CD14^+^, CD19^+^, CD56^+^, αβ, γδ, and NKT (CD3^+^/CD56^+^) cells were sorted from PBMCs using the cell sorter (MACS Quant^®^ Tyto^®^, Miltenyi Biotec, Bergisch Gladbach, Germany) with corresponding anti-human antibodies (sorting purity of subpopulations > 95%). Genomic DNA (gDNA) was isolated from peripheral blood cells and the different sorted populations with NucleoSpin Tissue kit (Macherey Nagel, Düren, Germany) following the manufacturer’s instructions.

### 2.3. Lymphocyte Percentage and IL-2RG Expression

Isolated PBMCs were stained with antibodies against human CD132 (IL-2RG), CD3, CD4, CD8, CD14, CD19, CD56, TCRαβ, and TCRγδ (Miltenyi Biotec, Bergisch Gladbach, Germany). Samples were incubated with the antibodies for 10min at room temperature in the dark and were read by flow cytometer (BD FACS Calibur, BD Biosciences, Franklin Lakes, USA). Data were carried out using FlowJo 10.7.2 software (BD Biosciences, Franklin Lakes, NJ, USA). 

### 2.4. Intracellular pSTAT5 Staining

PBMCs (1 × 10^6^ cells per sample) were stimulated with or without IL-2 (100, 1000, and 10,000 U/mL), IL-7 and IL-15 (1, 10, and 100 ng/mL) (Miltenyi Biotec) for 15 min at 37 °C, respectively. The cells were fixed and permeabilized using Cell Signaling Buffer Set A (Miltenyi Biotec) in accordance with the manufacturer’s instructions. After washing with EDTA buffer (0.5 mM, ThermoFisher, Waltham, MA, USA), the cells were stained with mouse anti-pSTAT5 (pY694) (BD Biosciences, Franklin Lakes, NJ, USA) and antibodies of CD4^+^, CD8^+^, and CD56^+^ (Miltenyi Biotec, Bergisch Gladbach, Germany) against cell surface antigens incubating at room temperature for 30 min. The samples were taken for flow cytometry acquisition and were analyzed using FlowJo 10.7.2 software (BD Biosciences, Franklin Lakes, NJ, USA).

### 2.5. Genetic Analysis

The gDNA of peripheral blood and sorted cells was used to amplify the region covering the *IL2RG* mutation with the pair of primers (Appendix A). Primers were designed online (https://www.ncbi.nlm.nih.gov/tools/primer-blast/) and synthesized by Metabion (Planegg/Steinkirchen, Germany). The PCR reaction was performed using GoTaq^®^ Green Master Mix (Promega, Madison, WI, USA) as follows: 95 °C for 2 min for initial denaturation, followed by 40 cycles of 95 °C for 40 s, 62 °C for 30 s, and 68 °C for 1 min. PCR products were purified by a QIAquick PCR purification kit (Qiagen, Dusseldorf, Germany). Then, 75 ng of PCR product and 4 µL of the forward primer (20 µM) were mixed with water in a final volume of 17 µL and samples were sent for Sanger sequencing analysis to Eurofins Genomics (Ebersberg, Germany). 

### 2.6. Maternal Engraftment

To exclude the persistence of maternal CD3^+^ T cells, chimerism analysis was performed by PCR amplification of short tandem repeat (STR) sequences. Several STR loci of maternal and patients’ gDNA from CD3^+^ T cell samples were amplified by Taq-Polymerase (Qiagen, Dusseldorf, Germany) with different primer pairs (Thermofisher, Waltham, MA, USA, sequences in Appendix A) to obtain at least one informative locus. Detection of patients and maternal DNA was carried out on sequential patient samples using 3130-16 Genetic Analyzer and GeneMapper Software (Thermofisher, Waltham, MA, USA).

### 2.7. TREC Quantification 

To study the T-cell thymic output, the quantification of T-cell receptor excision circles (TREC) [24] was performed with the X200 AutoDG Droplet Digital PCR System (Bio-Rad, Hercules, CA, USA). The sequences (Appendix A) of primers and probes for TREC and *RPP30* (reference gene) and protocol with ddPCR were previously published [25]. All primers and probes were synthesized by IDT (Coralville, IA, USA). 500ng of gDNA isolated from peripheral blood was added to mixing reaction with 2X no dUTP Supermix for probes (Bio-Rad, Hercules, CA, USA), uracil N-glycosylase (UNG, Thermofisher, Waltham, MA, USA), the primers (0.5 uM) and probes (0.15 uM). The droplets were generated after loading 20 µL of the reaction mixture and 70 µL of droplet generating oil (Bio-Rad, Hercules, CA, USA) into the DG8 kit (Bio-Rad, Hercules, CA, USA). Then, droplets were transferred into a 96 deep-well reaction plate with a multichannel pipette and the plate was sealed with a sealer (Bio-Rad, Hercules, CA, USA). Following settings of PCR was performed: 50 °C for 2 min (UNG digestion), 95 °C for 15 min, followed by 50 cycles of 95 °C for 30 s and 60 °C for 1 min and a final step at 98 °C for 10 min. The plate was read with the QX200 Droplet Reader (Bio-Rad, Hercules, CA, USA), and data were analyzed using QuantaSoft software version 1.7.4 (Bio-Rad, Hercules, CA, USA). TREC (copies/µl blood) was quantified as described before [25].

### 2.8. T-Cell Receptor Repertoire Analysis and CDR3 Spectra-Typing

PCR high-resolution spectra type analysis of peripheral human TCR repertoires was done as previously described [26]. Shortly, TCRβ sequences were first amplified with a panel of 27 Vβ-gene-specific 5′ primers (Vβ), and a C-gene-specific reverse primer. The amplification products were then fluorescently labeled using an additional 3′ C-gene primer and resulting run-off products identified by length polymorphism on polyacrylamide sequencing gels to reveal precise sizes of amplicons. Appropriate software was used to depict the size and number of peaks corresponding to discrete CDR3 lengths.

### 2.9. T-Cell Proliferation

Purified CD3^+^ T cells were cultured with TexMACS medium (Miltenyi Biotec, Bergisch Gladbach, Germany) with different supplements of IL-2 (50 U/mL), IL-7 (10 ng/mL) and IL-15 (5 ng/mL), as well as TransAct (Miltenyi Biotec, Bergisch Gladbach, Germany) in the plate at 37 °C with 5% CO_2_. Cell counting via flow cytometer (BD FACS Calibur, BD Biosciences, Franklin Lakes, USA) was performed on days 1, 3, 5, 7 and 9 and was analyzed using FlowJo 10.7.2 software (BD Biosciences, Franklin Lakes, USA). The proliferation of T cells was dependent on the ratio of the number of cells on the counting day to the first day. 

### 2.10. Statistical Analysis

The statistical analysis was performed using the Graph Pad Prism 9.1.2 software (GraphPad Software, San Diego, CA, USA). 2-way ANOVA was used in STAT5 phosphorylation and T cells proliferation analysis, whereas unpaired *t*-test was applied to the comparison of lymphocyte percentage and TREC quantification to determine significant differences (*p* < 0.05, * < 0.05, ** < 0.01, *** < 0.001, **** < 0.0001) of mean values between patients and healthy controls.

## 3. Results

### 3.1. Case Report

Three brothers were born to nonconsanguineous healthy German parents. Although the clinical symptoms varied in their severity, all of them showed signs of increased susceptibility to opportunistic respiratory and viral infections. Patient 1 (P1) was affected the most, suffering from molluscum contagiosum and cutaneous warts on both hands and feet since the age of 6 and 11, respectively. Before the exploration of a genetic mutation at the age of 19, P1 was treated with various therapies to reduce warts. No satisfying improvement has been achieved attempting all kinds of topical and systemic treatments, such as cryotherapy, keratolysis, curettage, laser therapy, imiquimod cream, local IL-2 injection and retinoids. Subcutaneous injection of interferon alpha (INF-α) was applied twice within four years, whereas the first therapy cycle showed more improvement by light regression of warts. However, both times the application had to be stopped due to severe side effects (e.g., recurrent fever, hair loss, mental health issues). In addition, the patient developed chronic respiratory infections leading to bronchiectasis after being hospitalized twice due to severe cases of pneumonia. Genetic examination excluded cystic fibrosis (CF) and primary ciliary dyskinesia (PCD). Therefore, some form of cellular immunodeficiency was firstly suspected in P1 at the age of 19 years. The immunological phenotype and genetic analysis (indicated in Section 3.2 and Section 3.3, respectively) indicated a late-onset combined immunodeficiency disorder, thus he was referred to our hospital and underwent a detailed immunological investigation, followed later by his mother and younger brothers. After confirmation of immunodeficiency, P1 received anti-microbial prophylaxis and immunoglobulin substitution therapy due to his symptomatic appearance. Since following this treatment plan, no severe bacterial infection has occurred, even though cutaneous warts and bronchiectasis are still present. Meanwhile, recalling the growth of Patient 2 (P2) and Patient 3 (P3), they also suffered from recurrent warts and molluscum contagiosum since their early childhood. The warts of P2 (25 years old) had resolved eventually without specific treatment at the age of 14, whereas P3 (23 years old) is still moderately affected. Moreover, P2 shows no signs of chronic respiratory infections either, whereas P3 was diagnosed with bronchiectasis and atelectasis (at the age of 21 years old) after hospitalization because of a severe case of pneumonia. Furthermore, antibiotic treatment was necessary multiple times due to recurrent respiratory tract infections in the past (Table 1).

### 3.2. Clinical and Immunological Phenotype 

P1 presented with generalized cutaneous warts on both hands and feet at first presentation (Figure 1a). Additionally, he was prone to recurrent respiratory infections, and therefore, a high-resolution computerized tomography (CT) scan was initiated. Here, the development of mild bronchiectasis was detected (Figure 1b). Furthermore, evaluation of immunophenotype revealed differed levels of immunoglobulin subclasses (Table 2) including slightly elevated IgG1 (1670 mg/dL), low IgG2 (79 mg/dL) and undetectable IgG4 (<0.3 mg/dL), respectively. IgA, IgM and IgE immunoglobulins were within the normal range. In addition, the analysis of absolute lymphocyte count (Figure 1c–h) demonstrated low CD4^+^ T-cells (Figure 1f) in P1. Moreover, we noticed massively decreased CD56^+^ NK cell numbers (Figure 1e) combined with a shifted CD4^+^/CD8^+^ ratio (<1.0) (Figure 1h). Overall, these data completed the T^low^B^+^NK^low^-phenotype and strengthened the suspicion of a cell-mediated immunity disorder. To confirm the numerical status of the patient’s lymphocytes, subpopulation analysis was done by flow cytometry in comparison with healthy controls. We observed reduced CD4^+^ and increased CD8^+^ T cells in patients (Figure 1i, Table 2). It was noted that the percentage of αβ and γδ T cells in peripheral CD3^+^ T cells were within the normal range (data not shown), while the percent of CD8^+^/TCR γδ T cells was above the normal range (Figure 1j).

### 3.3. Genetic Analysis 

The clinical and immunological manifestations of the three patients are similar but non-characteristic, therefore, genetic analysis for an inherited immunity disorder was initiated by NGS which was performed by the Human Genetic Center of Tübingen to analyze *JAK3, IL2RG, ILR7, PNP, ZAP70, RFXANK, CIITA, RFX5, FOXN1, RFXAP, CD3D, CD3E, STK4, CD3G, CD247, UNC119* and *CD8A* genes. Finally, a novel c.458T > C; p.Ile153Thr mutation in exon 4 of *IL2RG* was identified. The genetic finding was confirmed by Sanger sequencing of gDNA from whole blood samples of the patients (P1, P2, P3) (Figure 1k). 

### 3.4. Functional Analysis

Because of the mutation’s novelty, the analysis was performed primarily in terms of the expression and function of IL-2RG. Expression of IL-2RG protein (CD132) on the surface of CD3^+^, CD4^+^, CD8^+^, CD14^+^, CD19^+^ and CD56^+^ cells was detected with normal expression levels, compared with healthy controls (Figure 2a). As for the function of the mutated γ_C_, CD4^+^ and CD8^+^ T cells showed similar STAT5 phosphorylation to supraphysiological concentrations of IL-2 (1000 and 10 000 U/mL) compared to healthy controls. Interestingly, we observed a difference using lower concentrations for stimulation (100 U/mL). While healthy T cells showed almost full responsiveness (CD4^+^ T cells: 68.9% and CD8^+^ T cells: 63.1%), only a portion of the patients CD4^+^ and CD8^+^ T cells were stimulated (P1 36,6% and 22.0%; P2 54.8% and 23.0%) (*p* < 0.05 in CD8^+^ T cells) (Figure 2b). Additionally, whether to check if IL-7RG and IL-15RG signaling was affected, the titration experiment was repeated by stimulating the cells with recombinant human IL-7 and IL-15 (1, 10, and 100 ng/mL), respectively. However, no significant difference could be observed (Appendix A, b).

### 3.5. Sequencing of Multiple Subpopulations

The diffuse clinical and functional presentation of the patients prompted us to assume possible somatic mosaicism. Consequently, precise genetic analysis was performed by sequencing the gDNA of multiple immune cell subsets, which were compared with mother and healthy control CD3^+^ T cells sequence (Figure 3a,b). CD14^+^ monocytes, CD19^+^ B cells, and CD56^+^ NK cells showed the mutated *IL2RG* gene. However, some wild-type alleles were detectable in T cells subset of CD3^+^, CD4^+^, CD8^+^, αβ, γδ, CD3^+^/CD56^+^ (Figure 3c–e). Furthermore, in order to identify the source of wild-type alleles, short tandem repeat (STR) marker analysis (Figure 4a) of dissimilar peaks excluded the possibility of circulating maternal engrafted T cells. Therefore, we presumed that the wild-type alleles must be a result of “natural gene therapy” generated by spontaneous reversion leading to somatic mosaicism.

### 3.6. Quantification of T Cell Receptor Excision Circles and T Cell Proliferation

TREC levels in the peripheral blood of patients were much lower (<10 copies/µL) than healthy donors, indicating that the patients have low thymic output (*p* < 0.05; Figure 4b). This output is persistent, as the absolute number of T cells is stable in the long-term (Figure 1c,f,g). As for the proliferation in vitro, CD3^+^ T cells from patients did not respond towards IL-7 and IL-15 (*p* < 0.05 and *p* < 0.0001 on day 7 and day 9, respectively), but could respond to IL-2 in combination with TransAct in similar levels to the healthy controls (Figure 4c,d).

### 3.7. TCR Vβ Repertoire Analysis

The V(D)J rearrangement process generates CDR3 regions varying up to 30 amino acids in length. Amplicons of different lengths show a typical bell-shaped distribution of CDR3 sequences of different lengths with a maximum value of 12–15 amino acids in healthy adults. The CDR3 distribution patterns of all Vβ families of the three siblings with *IL2RG* mutation are dramatically skewed (Figure 4e). The highly significantly reduced TCR repertoire diversity is characterized by a significantly reduced complexity score. The complexity of Vβ- (and Vα-chain) repertoires can be determined by counting the number of peaks in spectra type analysis. A score of 9 describes a normal CDR3 size variability of 8–10 peaks per Gaussian curve, a score of 1 refers to Vβ family profiles showing a single peak, 0 describes the absence of peaks. The overall TCR complexity (complexity score) is the sum of 27 individual TCR Vβ- family scores with a maximum of 27 × 9 = 234 for the Vβ families. P1, P2, P3 showed the score of 64/243 (i.e., 0.26 of normal), 86/243 (i.e., 0.35 of normal), and 78/243 (i.e., 0.32 of normal), separately. 

## 4. Discussion

### 4.1. p.Ile153Thr Results in Hypomorphic Mutation for IL-2RG Signaling

In this study we report a novel missense *IL2RG* mutation (c.458T > C) in three brothers with an atypical X-SCID phenotype diagnosed with recurrent chronic infections (respiratory infections and HPV-associated cutaneous warts), slightly decreased CD4^+^ T cells and NK cell lymphocytes and abnormal immunoglobulins. This mutation results in the replacement of Isoleucine 153 by Threonine (p.Ile153Thr) in the extracellular domain and has not been described before. Our patients showed a mild phenotype with low-normal T and NK cells counts and peripheral blood lymphocyte subsets were able to express γ_C_ normally, even though STAT5 phosphorylation in CD4^+^ and CD8^+^ T cells was only partially detectable after stimulating with low concentration (100 U/mL) of cytokines of IL-2. In contrast, STAT5 phosphorylation of these cells could respond excellently to stimulation with high concentrations (1000 and 10,000 U/mL) and other cytokines, such as IL-7 and 15 (Appendix A, b). This might indicate that the IL-7R and IL-15R signaling pathways of the patients are intact and they are coping with a hypomorphic mutation for IL-2R signaling that can be restored if it is properly stimulated to a certain threshold. Since the STAT5 phosphorylation gradually becomes similar to healthy levels with the increase in IL-2 concentration, we hypothesize that the reason for the poor proliferation in vitro observed in the absence of supplemented IL-2 in the patients’ CD3^+^ T cells could be a result of failure to achieve IL-2 levels above this threshold. The T-cells’ own production of IL-2 is not sufficient to support their proliferation and their survival is compromised in the absence of exogenous IL-2. This is also consistent with the fact that the T-cells can proliferate when IL-2 and TransAct are provided to the medium simultaneously (Figure 4d).

### 4.2. p.Ile153Thr vs. p.Ile153Asn Cases Can Explain Different Outcomes of X-SCID

A similar point mutation (c.458T > A) affecting the same amino acid (p.Ile153Asn) was previously described by Puck et al. [1]. In a posterior publication from the same group, it was reported that the p.Ile153Asn substitution does not cause any impact on the expression of the receptor, but it negatively affects the IL-2 binding, resulting in lower ligand affinity [2]. These results fit with our observations for p.Ile153Thr mutation and made it relevant to compare the symptoms of patients carrying these missense mutations. To our knowledge, no detailed clinical report is available for p.Ile153Asn patients. Therefore, we assume it was reported as a classic case of X-SCID, which is not consistent with the mild phenotype observed in p.Ile153Thr. We found two crucial differences between both cases that could explain the different outcomes: Dissimilar impact in the protein structure: In order to further understand the differences between both mutations at the molecular level, we studied position 153 in the interaction with IL-2 (Appendix A). It is found in the extracellular domain of the receptor, more concretely in the cytokine binding surface [27]. The wild-type isoleucine at this position is part of a β-sheet motif. As a hydrophobic amino acid, its side chain is not very reactive but could be involved in ligand recognition [28]. The substitution for asparagine could have a severe consequence because of its polar nature, which could disrupt the β-sheet motif. This is not the case in the substitution for threonine since this slightly polar amino acid presents two non-hydrogen substituents attached to the beta carbon (similarly to isoleucine) which restricts the adoptable conformations from the main chain and could maintain the β-sheet conformation. Thus, we hypothesized that Ile153Thr mutation has a lower impact on the tertiary structure of IL2RG and would support a more benign outcome. It was also considered that Ile153Asn could be responsible for additional, aberrant modifications in the receptor, such as N-linked glycosylations, but the existence of proline in position 154 hampers the formation of a suitable N-glycosylation motif and makes this event very unlikely [29].The possible absence of natural reversion in p.Ile153Asn: The genetic analysis performed to identify this mutation was based on the abolishment of one SauIIIA restriction site in the mutant sample [2]. In the hypothetical presence of natural reverted cells, the restriction digestion would be possible and visible in the gel, which was not reported in the publication. This would suggest an absence of natural reverted cells, however, a genetic analysis using the genomic DNA from blood cells or PBMCs would hide a potential reversion in the lymphocyte population. A more detailed analysis of different subpopulations including T cells, B cells and NK cells would be needed to fully address this possibility.

### 4.3. Natural Reversion from P1, P2 and P3 Is Originated in Early Progenitor T-Cells

Regarding natural reversion in p.Ile153Thr patients, somatic mosaicism was found in T-cell subsets in further genetic studies of lymphocytes. It is worth highlighting that a simultaneous C > T reversion of a specific point mutation in the same T-cell subpopulations of three siblings is a statistically highly unlikely event, however, STR analysis excluded the possibility of maternal engraftment. It suggested a spontaneous somatic reversion happening because of natural gene therapy, that the event does not happen accidentally and is somehow determined by a genetically but yet unknown mechanism. According to the development of lymphocytes, we hypothesized that all reversal subpopulations in three brothers were derived from early revertant progenitor T cells, which was also reported previously [16,22]. In addition, the reversion was present not only in TCR αβ^+^ T cells but also in TCR γδ^+^ cells. The genetic reversion could lie between the CD4^−^CD8^−^ double-negative and CD4^+^CD8^+^ double-positive thymocyte stage and have occurred during or before a differentiation stage in the thymus that does not depend on CD132 expression, since the separation in TCR γδ^+^ and TCR αβ^+^ T cells takes place between the CD4^-^CD8^−^ double-negative and CD4^+^CD8^+^ double-positive thymocyte stage of T-cell differentiation in the thymus [30]. Furthermore, the reversion was not found in either monocyte, NK, or B cells, which implies that the revertant mutation most likely occurred after B cells and NK cells commitment or that it offered no survival advantage compared to subsets of T cells.

It can be widely observed that reverted mutations in the *IL2RG* gene can display within a large range of clinical and immunological findings. Patients with low T- and NK-cell counts are documented with T452C, A284-15G and T455C mutations [10,17,23], normal STAT5 phosphorylation is reported for a patient with T466C mutation [19], and even normal TCR repertoire in patients with A655T and T343C mutations [16,20]. Therefore, it is important to underline the differential diagnosis of SCID variants in terms of functional and clinical characterization beyond the infant period. 

### 4.4. Defect in IL-2RG Signaling Is Responsible for Inverted CD4^+^/CD8^+^ Ratio and Low Amounts of Naïve T Cells Compared to Memory T Cells

Our results show a reversion in all CD3^+^ T cells and an inverted CD4^+^/CD8^+^ ratio. Kuijpers et al. reported a similar case of a 6-years old boy with an inverted CD4^+^/CD8^+^ ratio [16]. According to their explanation, this could be generated by a slower cellular kinematic in CD4^+^ T cells in contrast to a higher antigen-driven expansion in CD8^+^ T cells, which could be a feasible explanation for the cases of this study. In addition, we observed increased CD3^+^CD8^+^CD28^−^CD27^−^ memory T cells and low CD3^+^CD4^+^CD45RA naïve T cells (Table 3). Interestingly, the STAT5 signaling pathway in CD8^+^ T-cells for all three patients can be activated by IL-7 and IL-15 in a similar way as healthy controls (Appendix A). In contrast to naïve T cells, memory T cells’ survival depends more on these cytokines than on IL-2 [31]. This is in line with a survival advantage of reactive/memory T cells versus naïve T cells.

### 4.5. Vβ Families and TCR Repertoire Are Skewed and Shared in P1, P2 and P3

Puzzlingly, the phenotypes of the three individuals are highly similar in Vβ families that are missing and those reduced to a single peak they are almost completely identical. Moreover, many families reduced to single peaks and highly overexpressed is highly suggestive for oligo or even monoclonal expansion as reported before [32,33]. We speculate that the skewed TCR repertoire may mainly consist of public TCRs that are virus specific. This also fits the finding that there are less naive, but more memory T cells in the T-cell compartment (Table 3). Both cytokines cannot be bound, however, IL-2 is generated when a T cell recognizes its cognate antigen. Hence, there is a survival advantage for reactive/memory T cells. A skewed repertoire may also be due to inefficient and/or abnormal generation of TCRs, reported earlier [34] which, however, was not the focus of the study. Taken together, the immunoscope of the peripheral T-cell pool of the three individuals reveals substantial gaps suggestive of a limited functional reactivity. Furthermore, low thymic output skewed TCR Vβ repertoire and poor proliferation of CD3^+^ T cells *in vitro* prompt us to consider carefully the possibility of T-cell exhaustion over time, though the numbers of both CD4^+^ and CD8^+^ T cells of P1 were steady and he showed no signs of clinical deterioration under the long-term immunological status for 7 years. Further follow-up is required to determine if the patients can maintain long-term T cell immunity. 

**Table 3 genes-13-00035-t003:** Surface marker analysis of peripheral blood mononuclear cells.

	P1 (%)		Reference (%) [35,36,37].
CD3^+^	**85.10**	+	55.00–83.00
CD4^+^	**21.90**	-	28.0–57.00
CD8^+^	**59.10**	+++	10.00–39.00
TCRαβ^+^	96.20		88.00–98.00
TCRγδ^+^	3.70		1.00–12.00
CD3^+^CD4^+^CD45RA^+^	**10.00**	--	21.00–58.00
CD3^+^CD4^+^CD45RO^+^	**85.90**	++	35.00–73.00
CD3^+^CD8^+^CD45RA^+^	**4.92**	---	23.00–73.00
CD3^+^CD8^+^CD45RA^−^	22.87		13.00–43.00
CD3^+^CD8^+^CD28^-^CD27^−^	**53.90**	++	1.60–36.00
CD19^+^	13.20		6.00–19.00
CD16^+^CD56^+^	**22**	---	90–600

Relative numbers of cells expressing surface markers are shown. Values different than reference are marked in bold.

### 4.6. Final Conclusions

Regarding the different clinical symptomatology between the studied patients and despite sharing the same mutation, we conclude that there are more factors involved in the development of their individual conditions (Table 4). Even though the reversion is observed in all three siblings, differences in their genetics, epigenetic profile and environmental conditions might explain the disparity between P1 symptoms and the rest of the brothers. We conclude that the reversion allows the restoration of T-cell function and immunity for at least 26 years of P1 who is the longest surviving case in comparison with atypical X-SCID patients reported with *IL2RG* reversion till now [10,15,16,17,18,19,20,21,22,23]. The precursor cells from the patients were not studied nor analyzed in this study due to ethical reasons. It is worth acknowledging that according to the advantages of revertant cells surviving and functioning in vivo of patients, gene correction via CRISPR/Cas9 or prime editing could provide a promising treatment for patients in the future if their condition deteriorates.

## Figures and Tables

**Figure 1 genes-13-00035-f001:**
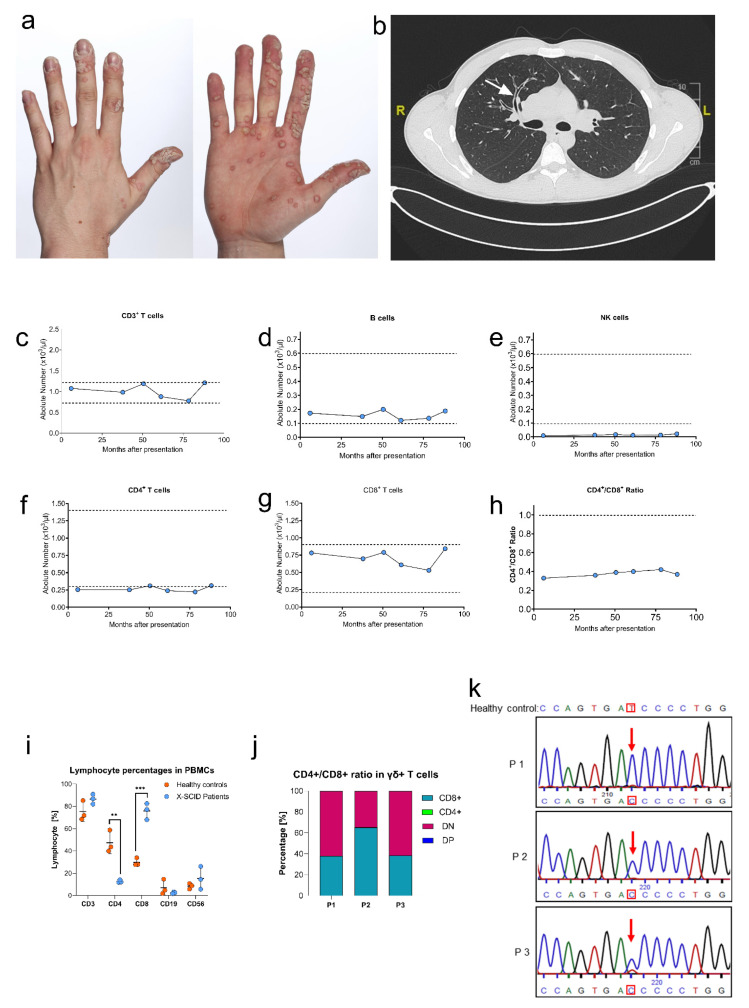
Clinical presentation of patient 1 (P1): (**a**) Hand warts; (**b**) Chest CT (19-year-old) with features of respiratory infections. Long-term immunological status of P1: absolute number of CD3^+^T cells (**c**), B cells (**d**), NK cells (**e**), CD4^+^ T cells (**f**), CD8^+^ T cells (**g**), and CD4^+^/CD8^+^ ratio (**h**). Dotted lines indicate upper and lower limits of normal for corresponding age, for CD4^+^/CD8^+^ ratio is inverted. Lymphocyte subpopulation ratios in PBMCs (**i**): The three patients had lower proportions of CD4^+^ T cells than healthy controls (*n* = 3, *p* ** < 0.01), while the percentage of CD8^+^ T cells were significantly higher than healthy controls (*p* *** < 0.001). CD4^+^/CD8^+^ ratio in γδ+-T cells (**j**): Compared with the reference range of double negative (DN) (>70%), CD8^+^ (30%) and CD4^+^ (<1%) in γδ T lymphocytes of human peripheral blood T cells, the percent of CD8^+^/TCR γδ T cells was above the normal range. Genetic analysis (**k**): DNA sequence analysis in whole blood from a healthy control and the three patients. The mutated nucleotide C replaced the wild-type T.

**Figure 2 genes-13-00035-f002:**
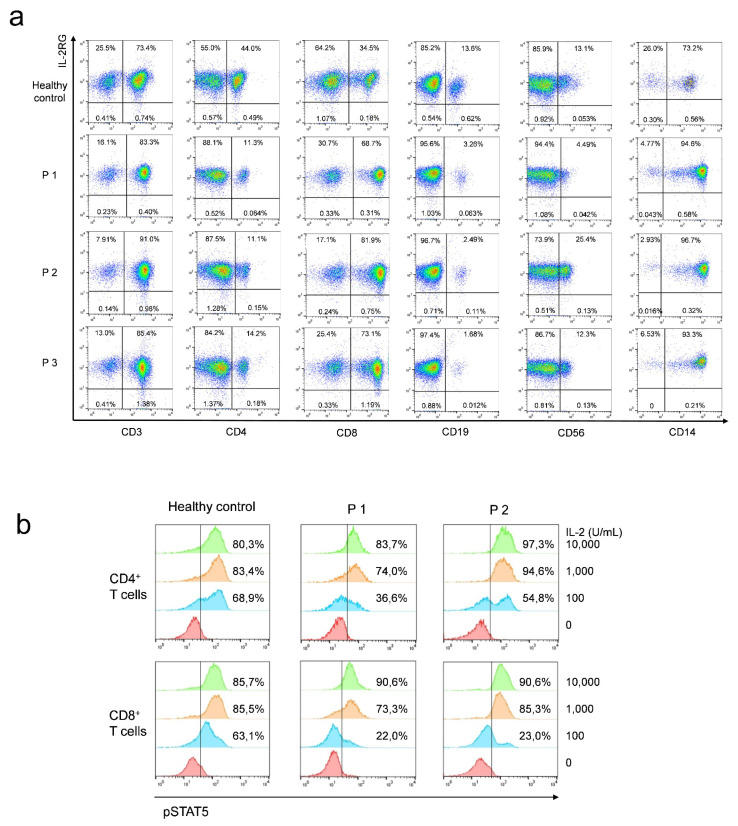
IL-2RG expression (**a**): Expression of IL-2RG was detected by flow cytometry. IL-2RG expression was normal in all three patients. PBMCs from the three patients and healthy controls (*n* = 3, only one healthy donor is shown) were stained with CD132-PE antibody and the corresponding lymphocyte subpopulation antibodies. STAT5 phosphorylation (**b**): CD4^+^ and CD8^+^ T cells of P1 and P2 exhibited partial γ_C_ signaling in response to stimulation with low concentrations of IL-2 (100 U/mL) compared to healthy donors (*p* < 0.05 in CD8^+^ T cells). High concentrations (1000 and 10,000 U/mL) showed similar STAT5 phosphorylation to healthy controls.

**Figure 3 genes-13-00035-f003:**
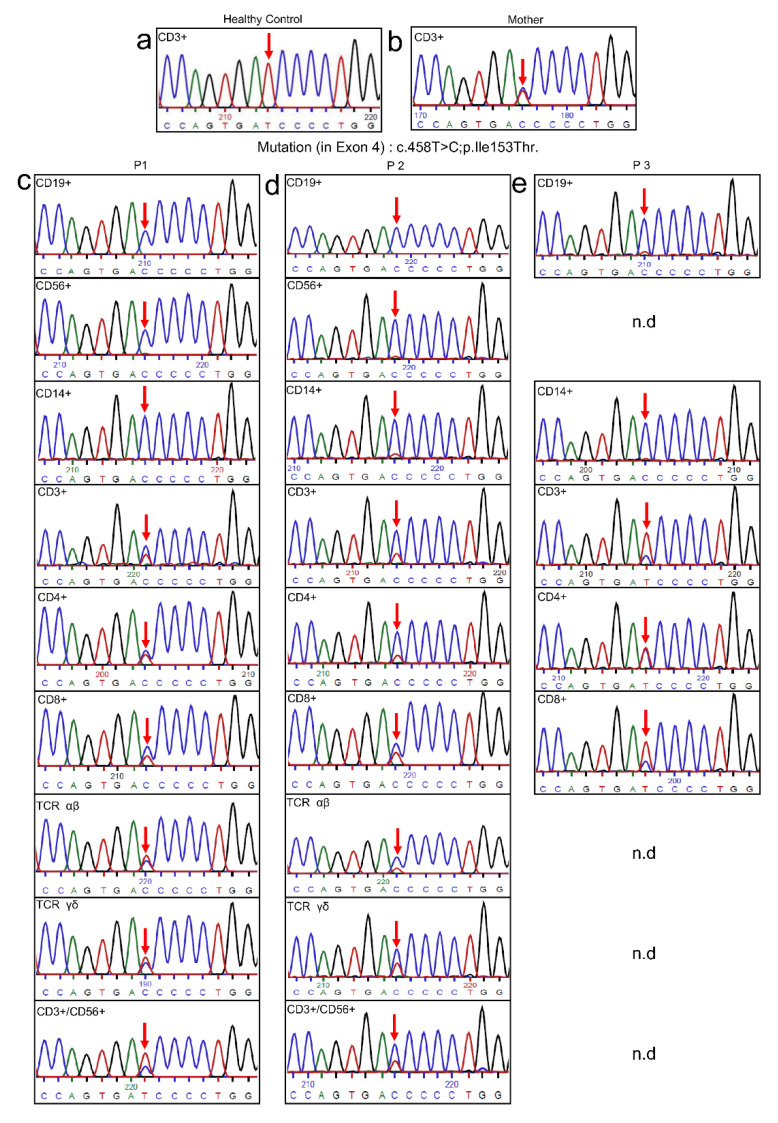
Gene sequence analysis was shown for CD3^+^ T cells of healthy control (**a**), CD3^+^ T cells of patients’ mother (which also presented the mutation) (**b**), and various lymphocyte subsets sorted from patients’ PBMCs (**c**,**d**,**e**): CD19^+^ B cells, CD56^+^ NK cells, CD14^+^ monocytes had the mutation, while CD3^+^, CD4^+^, CD8^+^, αβ, γδ T cells and NKT (CD3^+^/CD56^+^) cells have a reverted mutation (n.d = not determined due to lack of patient’s access).

**Figure 4 genes-13-00035-f004:**
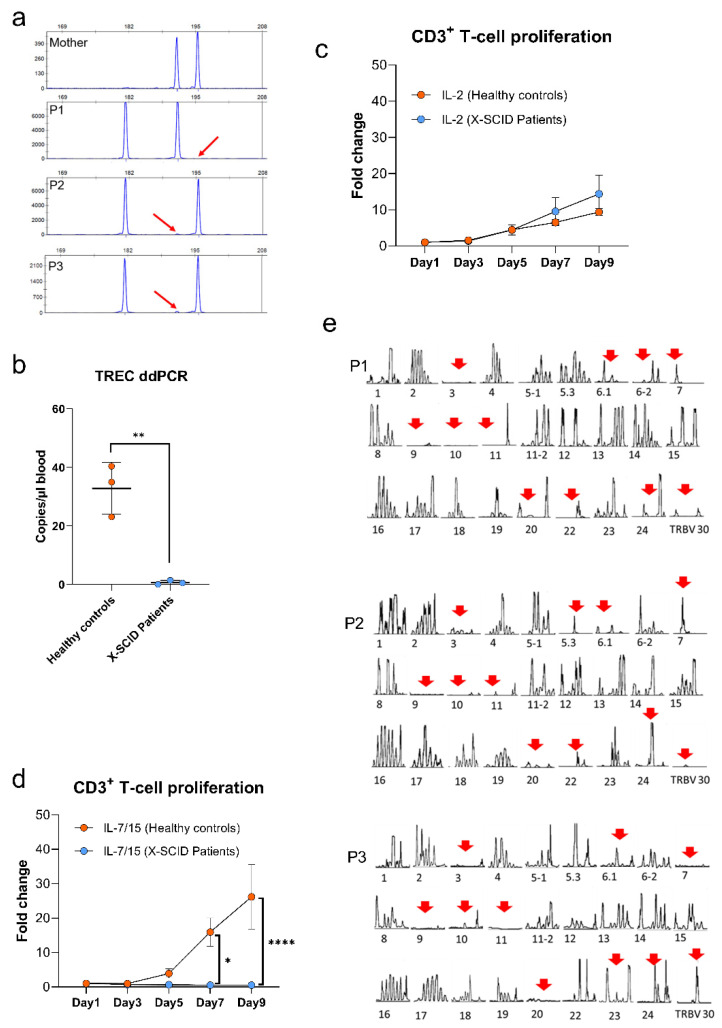
Maternal engraftment (**a**): Circulating maternal engrafted T cells were excluded due to dissimilar peaks. Typical STR profile with allele designation for genomic DNA extracted from CD3^+^ T cells of mother and three patients. The horizontal axis represents the values of fragment in base pairs and the vertical axis represents the values in relative fluorescent units (RFU). Amount of TREC (**b**) per µL of blood was determined by ddPCR. The amount of TREC (<10/µL blood) of three patients was much lower than healthy controls (*n* = 3) with a similar age (** *p* < 0.01). CD3^+^ T cells proliferation (**c**,**d**) response to IL-2 (50 units/mL), or IL-7 (10 ng/mL) and IL-15 (5 ng/mL), as well as TransAct (* *p* < 0.05, **** *p* < 0.0001). T-cell receptor repertoire analysis and CDR3 spectratyping (**e**): TCRβ sequences were first amplified with a panel of 27 Vβ-gene-specific 5′ primers (Vβ), and a C-gene-specific reverse primer. The amplification products were then fluorescently labeled using an additional 3′ C-gene primer and resulting run-off products identified by length polymorphism on polyacrylamide sequencing gels to reveal precise sizes of amplicons. The CDR3 distribution patterns of all Vβ families of the three siblings with a common gamma chain (γc)/interleukin-2 receptor gamma (IL-2Rγ) mutation is dramatically skewed. The highly significantly reduced TCR repertoire diversity is characterized by a significantly reduced complexity score. The complexity of Vβ- (and Vα-chain) repertoires can be determined by counting the number of peaks in spectratype analysis. A score of 9 describes a normal CDR3 size variability of 8–10 peaks per Gaussian curve, a score of 1 refers to Vβ family profiles showing a single peak, 0 describes the absence of peaks. The overall TCR complexity (complexity score) is the sum of 27 individual TCR Vβ- family scores with a maximum of 27 × 9 = 234 for the Vβ families. P1, P2, P3 showed the score of 64/243 (i.e., 0.26 of normal), 86/243 (i.e., 0.35 of normal), and 78/243 (i.e., 0.32 of normal), separately.

**Table 1 genes-13-00035-t001:** Clinical history of reported patients.

	P1	P2	P3
Patient (age, ethnicity)	26-year-old German male	25-year-old German male	23-year-old German male
Consanguinity		No	
Medical history	Viral infections with molluscum contagiosum and cutaneous warts due to *Human papilloma virus*, chronic respiratory infections, two hospitalization due to severe pneumonia within 7 years, *Pseudomonas aeruginosa*	Viral infections with molluscum contagiosum and cutaneous warts due to *Human papilloma virus*, otitis, susceptible to viral and fungi infections with skin rash	Viral infections with molluscum contagiosum and cutaneous warts due to *Human papilloma virus*, chronic respiratory infections with recurrent sinusitis, hospitalization due to a severe episode of pneumonia
Family history		No family history suggestive of immunodeficiency, mother detected as conductor of X-chromosomal recessive mutation	
Clinical phenotype (at diagnosis)	Bronchiectasis and generalized cutaneous warts	Asymptomatic	Bronchiectasis and right middle lobe atelectasis, local cutaneous warts
Viral examination	VZV IgG positive Positive EBV PCR Negative CMV PCR Positive HPV 2, 27, 57 PCR	n.a.	n.a.
Therapy	Local IL-2 injection Cryotherapy Laser therapy Keratolysis Imiquimod cream Retinoids INF-α IVIG substitution Antibiotic prophylaxis	-	Cryotherapy Laser therapy Keratolysis Imiquimod cream

VZV = varicella-zoster virus; CMV = cytomegalovirus; EBV = Epstein-Barr virus; HPV = *human-papilloma virus*; PCR = polymerase chain reaction; n.a. = not assessed.

**Table 2 genes-13-00035-t002:** General immunological characterization of the affected patients.

	P 1				P 2		P 3		Normal Range
	08/2019	01/2020	05/2020	07/2020	01/2020	05/2020	05/2020	07/2020	
Leukocytes, absolute/µL	5710	5790	6114	4610	5470	5200	4700	5300	3800–10,300
Lymphocytes, absolute/µL	1276	520↓	856↓	1011↓	1789	1456	1316	697↓	1100–3200
CD3+, absolute/µL (%)	866↓ (67.9)	n.a.	608↓ (71)	511↓ (50.5)	1502 (84)	1165 (80)	987 (75)	697↓ (47)	900–4500
CD4+, absolute/µL (%)	146↓ (11.4)	n.a.	146↓ (17)	151↓ (14.9)	272↓ (15.2)	233↓ (16)	197↓ (15)	178↓ (12)	500–2400
CD8+, absolute/µL (%)	685 (53.7)	n.a.	411 (48)	331 (32.7)	1073 (60)	815 (56)	737 (56)	430 (29)	300–1600
CD4+/CD8+ ratio	0.21		0.35	0.46	0.25	0.29	0.27	0.41	>1.0
CD19+, absolute/µL (%)	87↓ (6.82)	n.a.	103↓ (12)	76↓ (7.5)	179↓ (10)	175↓ (12)	132↓ (10)	119↓ (8)	200–2100
CD56+, absolute/µL	153 (12)	n.a.	21↓ (2.5)	23↓ (2.3)	72↓ (4)	22↓ (1.5)	61↓ (4.6)	163 (11)	100–1000
IgG, mg/dL	1760↑	1920↑	n.a.	1680↑	1550	1349	1470	n.a.	700–1600
IgG1, mg/dL	1670↑	1800↑	n.a.	n.a.	1550↑	n.a.	1270↑	n.a.	405–1011
IgG2, mg/dL	79↓	181	n.a.	n.a.	147↓	n.a.	168↓	n.a.	169–786
IgG3, mg/dL	63.1	73.1	n.a.	n.a.	149↑	n.a.	55.3	n.a.	11–85
IgG4, mg/dL	<0.3↓	10.9	n.a.	n.a.	0.6↓	n.a.	0.4↓	n.a.	3.0–201
IgA, mg/dL	240	269	n.a.	n.a.	230	n.a.	n.a.	n.a.	70–400
IgM, mg/dL	111	114	n.a.	n.a.	166	n.a.	n.a.	n.a.	40–230
IgE, mg/dL	16	19	n.a.	n.a.	9	n.a.	n.a.	n.a.	0–100

**Table 4 genes-13-00035-t004:** Characteristics of reported patients carrying the *IL2RG* c.458T > C; p.Ile153Thr hypomorphic mutation.

	P1	P2	P3
Thymic shadows		n.a.	
Blood count	CD4^+^ and NK cell lymphopenia	Low CD4^+^ T cells	Low CD4^+^ T cells
Immunophenotype		T^low^B^+^NK^low^	
γδT cells		Normal	
Extended immunophenotype		Low CD4/CD8 ratio < 1.0	
Immunoglobulin levels		Dysgammaglobulinemia	
Lymphocyte proliferation		Variable	
TRECs		Reduced	
TCR Vβ repertoire		Skewed	
Genetic findings	*IL2RG* c.458T > C; p.Ile153Thr		
IL-2RG expression		Normal	
STAT-5 Phosphorylation	Partially defective (FC)	Partially defective (FC)	n.a.

FC = flow cytometry; NK = natural killer; TCR = T cell receptor; TRECs = T cell receptor excision circles; n.a. = not assessed.

## Data Availability

The data presented in this study is available from the corresponding author upon reasonable request.

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
