# Peer review of "Somatic Reversion of a Novel IL2RG Mutation Resulting in Atypical X-Linked Combined Immunodeficiency"

_genes, 2021, doi:10.3390/genes13010035_

Round 1

Reviewer 1 Report

The current paper is a novel report on 3 patients who demonstrate varied symptoms of X-linked SCID. The authors report a case of spontaneous somatic reversion which allowed for one of the patients (P2) to have a milder disease over time. While this is not the first reported case, the authors identify a novel mutation in IL2RG and therefore add to the repertoire of diagnostic mutations that can allow for more personalized clinical outcomes and therapy. Therefore, this study is a valuable addition to the field. However, the following are some of the comments and concerns that need attention:

  1. Line 132-133 – TERC is introduced here for the first time and the full form needs to be spelled out. Further, a reference here that describes the basis is needed to direct readers.

  1. In results 3.2 the clinical details of all 3 patients are listed but the ratios and absolute numbers of (Figure 1 c-h) only patient 1 are shown. While all 3-patient lymphocyte % is shown in figure 1 I. This data is the main figure of the paper that will add to the impact.

  1. Table 1 has important details but is more suitable for supplemental data.

  1. Throughout the paper statistical significance is not mentioned (except for figure 1i). While a sample number is a deterrent and hence perhaps none are significant, it will still be helpful to know the p values. Furthermore, the authors can comment on the probability of their difference in observed ratios and percentages of lymphocytes being of biological significance. The lack of statistical analysis is very apparent in figure 2 and needs to be addressed.

  1. In 4.2, the authors focus on another study by Puck et.al. and draw conclusions based on their study to hypothesize on their own findings on the novel IL2RG mutations. They mention that they have studied the consequences of the mutation in affecting interaction with IL2. How was this done? This needs to be included in the main paper.

  1. In 4.3, the authors hypothesize how they support the clonal expansion of a single progenitor T cell. While they have explained their rationale in detail, without accompanying references, this section appears to be overreaching. Suitable references need to be included. Especially, regarding the fact that spontaneous somatic reversion is determined by a genetic yet unknown mechanism.

  1. Some of the tables are cut off and the column titles of column 2 in table 4 are missing or have shifted.

8. fig1 supplemental has not been quantified and I am confused with supplemental figure 2. If they have used a bioinformatics tool to predict interactions, then providing that explanation will be really helpful and make the paper stronger.

Overall, this paper set the ground for further investigation into the novel IL2RG mutation, and as the authors mention these findings argue for genetic evaluation of patients over time and not only at infancy.

Reviewer 2 Report

Congratulation for this extensive work! There are several suggestions:

Line 16 (in abstract) - replace ‘identified’ with ‘has been proven’ ..

Line  104: to correct the phrase ‘were treated with or without IL-2 (100, 1000, and 10 104 000 U/ml)...’

Line 120: Please explain why the amplicons were mixed with the forward primer (20μM) ..

In table 3: to correct   molluscum contaginosa  with molluscum contagiosum          

In table 3: to replace case with episode: severe case epidode of pneumonia
